# The Use of Voriconazole as Primary Prophylaxis for Invasive Fungal Infections in Patients Undergoing Allogeneic Stem Cell Transplantation: A Single Center’s Experience

**DOI:** 10.3390/jof7110925

**Published:** 2021-10-31

**Authors:** Ali Atoui, Nadine Omeirat, Omar Fakhreddine, Raquelle El Alam, Zeina Kanafani, Iman Abou Dalle, Ali Bazarbachi, Jean El-Cheikh, Souha S. Kanj

**Affiliations:** 1Bone Marrow Transplantation Program, Department of Internal Medicine, American University of Beirut, Beirut P.O. Box 113-6044, Lebanon; aa326@aub.edu.lb (A.A.); ia41@aub.edu.lb (I.A.D.); bazarbac@aub.edu.lb (A.B.); 2Department of Internal Medicine, American University of Beirut, Beirut P.O. Box 113-6044, Lebanon; no12@aub.edu.lb (N.O.); of06@aub.edu.lb (O.F.); 3Department of Diagnostic Radiology, American University of Beirut, Beirut P.O. Box 113-6044, Lebanon; ra314@aub.edu.lb; 4Infectious Disease, Department of Internal Medicine, American University of Beirut, Beirut P.O. Box 113-6044, Lebanon; zk10@aub.edu.lb

**Keywords:** hematological malignancies, allogeneic transplant, invasive fungal infection, primary prophylaxis, voriconazole

## Abstract

Background: Invasive fungal infections (IFI) following allogeneic stem cell transplant (allo-HCT) are associated with high morbidity and mortality. Primary prophylaxis using voriconazole has been shown to decrease the incidence of IFI. Methods: We conducted a retrospective analysis at the Bone Marrow Transplant (BMT) unit of the American University of Beirut including 195 patients who underwent allo-HCT for hematological malignancies and received voriconazole as primary prophylaxis for IFI. The primary endpoints were based on the incidence of IFI at day 100 and day 180, and the secondary endpoint based on fungal-free survival. Results: For the study, 195 patients who underwent allo-HCT between January 2015 and March 2021 were included. The median age at transplant was 43 years. Of the patients, 63% were male, and the majority of patients were diagnosed with acute myeloid leukemia (AML) (60%). Voriconazole was given for a median of 90 days and was interrupted in 20 patients. The majority of IFI cases were probable invasive aspergillosis (8%). The incidence of IFI including proven, probable and possible IFI was 34%. The incidence of proven and probable IFI was 5% were 8%, respectively. The incidence of proven-probable (PP-IFI) was 5.1% at day 100 and 6.6% at day 180. The majority of PP-IFI cases were invasive aspergillosis (8%). A univariate analysis of patients, transplant characteristics and IFI showed a significant correlation between the type of donor, disease status before transplant, graft-versus-host disease prophylaxis used and incidence of IFI. Only disease status post-transplant showed a significant correlation with fungal-free survival in the multivariate analysis. Conclusion: Primary prophylaxis with voriconazole in allo-HCT is associated with a low incidence of IFI. More studies are required to compare various antifungal agents in this setting.

## 1. Introduction

Invasive fungal infections (IFI) following allogeneic hematopoietic stem cell transplantation (allo-HCT) are associated with high morbidity and mortality. Based on several studies, the incidence of proven and probable IFI following allo-HCT (based on the revised European Organization for Research and Treatment of Cancer (EORTC) and the Mycoses Study Group (MSG) definitions) ranges from 8% to 12% in the first year [1]. Pathogens causing IFI are separated into three groups: *Candida* species, *Aspergillus* species and other molds including *Zygomycetes* and *Fusarium* species. Several international societies have published guidelines for prophylaxis of IFI in this patient population. In general, fluconazole, Itraconazole and voriconazole are recommended for prophylaxis of IFI in allo-HCT with no additional risk factors, with posaconazole recommended for patients with graft-versus-host disease (GVHD). For patients at high risk for IFI, primary prophylaxis with triazoles such as itraconazole, posaconazole, or voriconazole is recommended [2,3]. Studies have investigated the use of voriconazole in this setting. A study comparing prophylactic fluconazole with voriconazole showed no difference in the primary endpoint of fungal-free survival at 180 days. However, fewer IFI were observed with voriconazole in a subset of patients receiving allo-HCT for acute myeloid leukemia (AML) [4]. Another trial comparing voriconazole and itraconazole following allo-HCT for 100–180 days showed the superiority of voriconazole in the primary endpoint after incorporating survival without proven/probable IFI at 180 days and the ability to tolerate the drug for 100 days, with less than 14 days interruption [5]. Based on recommendations from the European Conference on Infections in Leukemia (ECIL-5) and the Italian Group for Bone Marrow Transplantation (GITMO), voriconazole prophylaxis is recommended as B-I evidence for both low- and high-risk patients [6]. The Infection Disease Society of America (IDSA) strongly recommends the use of voriconazole for patients with high risk of invasive aspergillosis [7].

Despite improvements in prophylaxis, IFI-related morbidity and mortality levels are still considered a substantial clinical burden, and only two randomized trials have investigated the role of voriconazole in this population. In this study, we evaluated the efficacy of prophylactic voriconazole in allo-HCT for the prevention of IFI at a tertiary medical center in Lebanon. We assessed the incidence of probable/proven IFI, survival without probable/proven IFI and the tolerability of voriconazole. 

## 2. Methods

### 2.1. Study Population and Data Collection

All adult patients who underwent allo-HCT at the Bone Marrow Transplant Unit of the American University of Beirut Medical Center (AUBMC) from January 2015 to March 2021 and received voriconazole for primary IFI prophylaxis were enrolled. The medical charts were reviewed retrospectively, and follow-up data including mortality was obtained in July 2021. Data were collected for each patient’s demographic, hematological disease and transplant characteristics including donor type, conditioning regimen and GvHD prophylaxis, post-transplant follow-up, incidence of proven and probable IFI and survival without probable or proven IFI. All patients received voriconazole from day −1 before transplant to day 100 post-transplant or until stopping immunosuppression. It was given at a dose of 4 mg/kg intravenously every 12 h during their hospital stay and 200 mg orally twice daily thereafter. Therapeutic drug monitoring was not checked during prophylaxis. 

The study received approval by the Institutional Review Board of AUBMC. All procedures performed were in accordance with the ethical standards of the 1964 Helsinki declaration. 

### 2.2. Diagnostic Procedures and Screening for IFI

All patients were screened for weekly serum galactomannan levels in 2015, and only in case of high suspicion of IFI from 2016 to 2021. In the case of fever, the standard procedures included blood cultures at 2 different time points, urine analysis and culture, chest X-ray, and in case of infiltrates or worsening respiratory status, were followed up with a computed tomography (CT) scan of the lungs. In the case that bronchoscopy was performed, fungal culture and galactomannan were requested from the broncho-alvelolar lavage (BAL) fluid. For patients with neurological symptoms, magnetic resonance imaging (MRI) of the brain and lumbar puncture were performed. In our study, all imaging for patients with confirmed proven and probable IFI were independently and blindly reviewed by a radiologist at our institution. 

### 2.3. IFI Classification and Endpoints

The EORTC/MSGERC consensus definitions of invasive fungal diseases (IFDs) were last revised and updated in 2019 [8]. IFI are classified into proven, probable and possible based on host factors, clinical features and mycological evidence. These definitions are not intended to guide or direct patient care; however, their original aim was to better define IFI for studies including patients with cancer and recipients of hematopoietic stem cell transplant (HCT) or solid organ transplant [9]. The primary endpoint of this study was based on the incidence of proven and probable IFI at day 100 and day 180 of HCT. Secondary endpoints were based on the determination of the risk factors for IFI after HCT and overall survival without proven or probable IFI. 

## 3. Statistical Analysis

Data were entered and analyzed using SPSS version 24.0 (IBM, Armonk, NY, USA). A two-sided statistical significance was set at a *p*-value of 0.05. The univariate associations were computed using Fisher’s exact test, bivariate Pearson correlation and Kruskal Wallis or Mann Whitney U tests, as appropriate.

## 4. Results

### 4.1. Patients and Transplant Characteristics

For the study, 195 adult patients with hematological malignancies who underwent allo-HCT between January 2015 and March 2021 were included. The median age at transplant was 43 years (IQR 16–75). Of the patients, 123 (63%) were male. The majority of patients (115 (59%)) were diagnosed with AML. Of the patients, 118 (60%) were in complete remission at the time of the transplant. All patients received peripheral stem cell source; of the patients, 113 (58%) had a matched related donor, and 82 (42%) had a haplo-identical mismatched related donor. Myeloablative conditioning was given as a treatment for 153 (78%) of patients. The post-transplant course was complicated by acute graft-versus-host disease (aGvHD) in 50 patients (26%). Patients with a high risk of IFI were as follows: 14 patients (7%) with grade III-IV aGvHD, 90 patients (46%) with CMV reactivation and 19 patients (10%) with refractory disease at the time of transplant. Table 1 and Table 2 summarize patients characteristics and post-transpant complications. Respectively. The median time of voriconazole prophylaxis was 90 days (range 80–100). Voriconazole was interrupted during the first 30 days post-transplant in 20 patients for elevated liver enzymes, one for seizures, one for visual hallucinations and one for severe nausea. Of these 20 patients, 16 patients received anidulafungin 100 mg daily. Only two patients had IFI (one proven mucormycosis and one probable pulmonary aspergillosis). The median follow-up time period was 14.7 months. 

### 4.2. Incidence and Characteristics of IFI

During the study period, the total number of patients who experienced possible, probable or proven IFI at any time post-transplant was 67 (34%). The number of patients with PP (proven-probable) IFI at any time post-transplant was 25 (12.8%), 10 of them during the first 100 days and 13 during the first 180 days after HCT. The day + 100 and day + 180 incidences of PP-IFI ere 5.1% and 6.6%, respectively. Among PP-IFI, invasive aspergillosis was the predominant infection for 16 cases (8.2%) of pulmonary aspergillosis at any time post-transplant. Of these cases, the incidences of pulmonary aspergillosis were 3% and 4.1% at day + 100 and day + 180, respectively. The second most common PP-IFI were invasive candidiasis in four patients and mucormycosis in four patients. 

Pathogens of proven IFI included *Candida glabrata* (*n* = 2), *Candida auris* (*n* = 1), one *Candida kefyr* (*n* = 1), and one case of cryptococcosis based on MRI brain and the presence of Cryptococcal antigen in the cerebrospinal fluid. Pathogens of probable IFI were *Aspergillus* species in three cases. The additional 13 cases of invasive aspergillosis were based on positive galactomannan in BAL or serum, in addition to CT scan findings (four showing dense, well-circumscribed lesions with or without halo sign and nine showing wedge-shaped and segmental or lobar pneumonia). There was no evidence of air crescent sign or cavities in any patient. Historically, the minimum inhibitory concentration (MIC) of voriconazole to Candida spp. were done only upon request. Therefore, MIC was only determined in one patient with *Candida glabrata* candidemia and revealed susceptibility to voriconazole. (MIC ≤ 0.12 ug/mL). 

The median time from HCT to the diagnosis of IFI before 180 days was 40 days (range, 11–165). For patients with proven mucormycosis, only one patient with refractory disease at the time of transplant experienced infection within 180 days of allo-HCT. Table 3 summarizes the characteristics of IFI. 

### 4.3. Risk Factors of IFI and Survival

According to a univariate analysis of clinical features including patients, disease, transplant characteristics and the incidence of IFI including proven, probable and possible IFI at any time after the transplant, it was found that the disease status before transplant, type of donor and type of GvHD prophylaxis correlated significantly with the incidence of IFI. The IFI rate was lower in patients who had complete remission before HCT than in patients with stable or refractory disease (30% vs. 55% vs. 58%, *p* = 0.023), in matched related donor stem cell compared to mismatched related donor (MMRD) recipients (23% vs. 48%, *p* = 0.0003) and in patients who did not receive post-transplant cyclophosphamide (PTCy) for GvHD prophylaxis (Cyclosporine (CsA): 24% vs. CsA and Mycofphenolate Mofetil (MMF): 24% vs. CsA, MMF and PTCy: 48%, *p* = 0.003). Table 4. Among 11 patients who underwent allo-HCT from MMRD and developed grade II-IV aGvHD, only 1 patient had IFI. 

The median overall survival without proven or probable IFI was not reached. Overall survival without proven or possible IFI at 100 and 180 days after HCT were 95% and 90%, respectively (Figure 1).

A Cox regression model was used to analyze factors affecting the chance of survival without proven or probable IFI, including age, sex, hematological diagnosis, type of donor, type of conditioning, the presence of aGvHD and disease status before HCT. Only the disease status before transplant was correlated with overall survival (OS) in patients with proven or probable IFI with a significantly higher OS in patients with complete remission compared to those with stable and refractory disease (*p* < 0.0001).

## 5. Discussion

In our study, we analyzed the efficacy of prophylactic voriconazole in allo-HCT recipients and found incidences of IFI of 5.1% and 6.6% at 100 and 180 days after HCT, respectively. The IFI incidence in our study was lower than results reported in previous studies evaluating the incidence of IFI in this population. In a study carried out by the Italian Group of Bone Marrow Transplantation (GITMO), the cumulative incidences of proven and probable IFI were 6.7% at 100 days and 8.8% at 12 months [1]. Another study from 11 Italian transplantation centers reported a 5-year incidence of 7.8% for IFI among more than 1200 allogeneic HCT recipients. Among the allo-HCT patients, significant risk factors included MMR in 14% of patients. The majority of invasive mold infections were caused by *Aspergillus* spp. in 94% of cases, while *Candida* spp. were the only yeast infections observed [10]. Mold infections in our study were caused by *Aspergillus* spp. in around 80% of cases, while *Candida* spp. were the only yeast identified. 

Mucormycosis has been reported as a breakthrough infection in patients on voriconazole from the MD Anderson Cancer Center [11]. Since voriconazole has no activity against mucorales, this has become a concern for centers witnessing an increased number of cases of infection with mucorales. We recently reported an increasing number of mucormycosis cases at AUBMC over the years, although not exclusively in allo-HCT patients [12]. Interestingly, in this study, we only observed one case of mucormycosis within 180 days after transplant, in a patient with refractory disease.

In our study, the incidence of invasive aspergillosis at any time after transplant was 8.2% compared to 2% of invasive candidiasis. Over the last decade, invasive aspergillosis has replaced invasive candidiasis as the most common fungal pathogens affecting allo-HCT recipients, with an estimated incidence of 6–7% in the first year following HCT compared to 1 to 5% of invasive candidiasis [10]. However, the incidence of invasive aspergillosis during the first 100 days after transplant was only 3%, suggesting that voriconazole prophylaxis (given for a median of 90 days) was quite effective in reducing the incidence of invasive aspergillosis. It is important to note that data on aspergillus susceptibility and speciation is lacking for Lebanon and most of the Arab countries. A recent review describes the epidemiology and calls for collaboration to better understand the epidemiology of resistant aspergillosis in the region [13]. It was only around two years ago we started referring our *Apsergillus* spp. to Austria (the laboratory of Professor Cornelia Lass-Floerl) for speciation and susceptibility testing. Therefore, it is not clear if the aspergillus infections seen in our patients in this series were due to intrinsically azole-resistant *Aspergillus* sp. or species that are traditionally susceptible and have acquired resistance after voriconazole exposure. Future data will hopefully elucidate this.

Multiple factors have been identified as risk factors for IFI during neutropenia in the transplant population, including older age, iron overload and prior IFI or colonization [14]. The GITMO has identified risk factors to indicate high risk for IFI during the early and late phase post-allo-HCT. Among these is the presence of grade III-IV aGvHD [6]. In our study, the presence of aGvHD did not correlate significantly with the incidence of IFI. However, a significantly increased incidence of IFI was observed in patients who received stem cells from MMRD. In a study published by Sun et al., the incidence of IFI was significantly higher after haploidentical HCT than that after HLA-matched HCT (7.1% vs. 3.3%, respectively; *p* = 0.007). Other factors associated with and increased risk of IFI in our cohort were disease status before HCT and the GvHD prophylaxis used. These results can be explained by prolonged neutropenia in patients with refractory and stable disease before transplant, in addition to the more severe immunosuppression with combination therapy of immunosuppressant agents for GvHD prophylaxis [15]. The emergence of reduced-intensity conditioning (RIC) has improved outcomes of HCT in patients with a high risk of transplant-related mortality (TRM) and has been widely used to reach around 50% of all allo-HCT recipients [16]. However, despite the favorable toxicity outcomes observed with RIC compared to myeloablative conditioning, the incidence of IFI was not significantly reduced using RIC [17].

Several international societies have published recommendations for prophylaxis of IFI in allo-HCT recipients [6,18,19]. Fluconazole, itraconazole and voriconazole are recommended for prophylaxis of IFI in these patients based on risk factors, taking into consideration the local epidemiology of mold/yeast infections to select the prophylactic agent. In a systematic review and meta-analysis of antifungal prophylaxis for IFI after allo-HCT recipients, systemic antifungal agents were associated with improved survival compared to placebo, no treatment or non-systemic therapy (relative risk: 0.62, 95% CI (0.45–0.85)). However, the study did not include new generation triazoles such as voriconazole. Voriconazole was compared to other antifungals in this setting in other reports. In a double-blind, randomized trial comparing prophylactic voriconazole with fluconazole, there was no difference in fungal-free survival at 180 days (hazard ratio: 1.07, 95% CI (0.82–1.4)). However, in a subgroup of patients who received allo-HCT, fewer IFI were observed with voriconazole than with fluconazole (hazard ratio (95% CI): 2.03 (1.29–3.19)). In a mixed treatment comparison study which allowed comparison of the use of two antifungals by assessing them relative to common comparator interventions between trials has shown superiority of voriconazole in reducing the incidence of IFI within 180 days post HCT with an odds ratio (95%) of 0.46 (0.28–0.73), 0.56 (0.28–0.73) and 0.52 (0.35–0.76) relative to fluconazole, posaconazole and itraconazole, respectively [20].

Fungal-free survival (FFS) at 180 days was around 90% in our study. A randomized, double-blind trial comparing fluconazole versus voriconazole for prevention of IFI in HCT recipients showed an FFS of 75% with voriconazole [4]. In a randomized trial comparing voriconazole and itraconazole for a primary objective of survival without PP-IFI at 180 days, there was no difference in FFS at day 180 between voriconazole (81.9%) and Itraconazole (80.9%). In a systematic review and meta-analysis including studies comparing different antifungal prophylaxis in patients with hematological disease, voriconazole was found to be the best prophylaxis options in the subgroup of patients who underwent HCT, especially allo-HCT [21]. 

Voriconazole was discontinued in 10% of the patients in our study due to elevated liver enzymes. In a retrospective study describing the clinical experience of primary prophylaxis with voriconazole in allo-HCT, early drug discontinuation occurred in 22.3% of patients due to hepatotoxicity [22]. To better investigate the role of therapeutic drug monitoring in this setting, a recent study investigated serum voriconazole levels in 151 patients who underwent allo-HCT. Although one-third of the patients achieved a subtherapeutic level, there was no correlation between drug level and risk of adverse events requiring discontinuation as well as breakthrough IFI [23].

## 6. Limitations and Strengths 

This is a retrospective study and had some limitations due to its design. The most significant strength of the study is the close follow-up period of each patient during the first year after allo-HCT. This allowed for a better evaluation of the incidence of IFI. 

## 7. Conclusions

In our study, the incidences of proven and probable IFI in allo-HCT patients using voriconazole prophylaxis were 5 and 8%, respectively. Current evidence-based guidelines recommend the use of antifungal prophylaxis for patients receiving allo-HCT. Voriconazole, a broad spectrum triazole, has demonstrated equivalent clinical efficacy to other older antifungal agents, which was confirmed in our study and is recommended for patients at high risk of IFI. Interruption and discontinuation of voriconazole was necessary in a subset of patients. New triazoles, including posaconazole and isavuconazole, have also been shown to be efficacious and safe in this patient population. Future studies are required to better compare various antifungal agents and their activity according to patients’ primary disease, risk factors and comorbid conditions to determine a precision medicine approach for the selection of the best agent according to efficacy and tolerability in order to improve outcomes.

## Figures and Tables

**Figure 1 jof-07-00925-f001:**
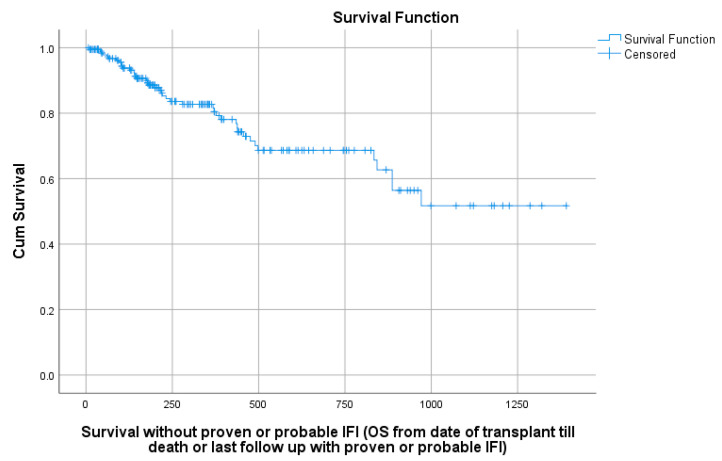
Kaplan–Meier curve of overall survival without proven or probable IFI.

**Table 1 jof-07-00925-t001:** Patients and disease characteristics.

Patients Characteristics	Study Population (*n* = 195)
Age at transplant, median (range)	43 (16–75)
Sex	
Female	72 (36%)
Male	123 (64%)
Hematological Disease	
AML	115 (59%)
ALL	29 (15%)
Lymphoma	31 (15%)
Other	20 (10%)
Number of lines before HCT, median (range)	2 (1–9)
Disease status before HCT	
CR	118 (60%)
SD	9 (5%)
Refractory	19 (10%)
Other	33 (17%)
Donor type	
MMRD	82 (42%)
MRD	113 (58%)
Conditioning type	
MAC	153 (79%)
RIC	42 (21%)
ATG	170 (81%)
aGvHD prophylaxis	
CsA	94 (48%)
CsA/MMF	17 (9%)
CsA/MMF/PTCy	84 (43%)
Year of HCT	
2015	21 (11%)
2016	36 (18%)
2017	32 (16%)
2018	42 (22%)
2019	25 (13%)
2020	30 (15%)
2021	9 (5%)
Time to ANC engraftment, median (range)	14 (8–31)

Abbreviations: AML: acute myeloid leukemia, ALL: acute lymphocytic leukemia, HCT: hematopoietic stem cell transplant, CR: complete remission, SD: stable disease, MMRD: mismatched related donor, MRD: matched related donor, MAC: myeloablative conditioning, RIC: reduced intensity conditioning, aGvHD: acute graft-versus-host disease, CsA: cyclosporine, MMF: mycophenolate mofetil, PTCy: post-transplant cyclophosphamide, ANC: absolute neutrophils count.

**Table 2 jof-07-00925-t002:** Post-transplant complications.

Post-Transplant Complications	Study Population (*n* = 195)
Acute GvHD	50 (27%)
Grade I	15 (8%)
Grade II-III-IV	28 (14%)
GI GvHD	18 (9%)
Skin GvHD	43 (22%)
GvHD systemic treatment	35 (18%)
Viral reactivation post–HCT	
CMV	90 (46%)
EBV	35 (18%)
BK Virus	33 (17%)

Abbreviations: CMV: Cytomegalovirus, EBV: Epstein Barr Virus, GI: Gastrointestinal.

**Table 3 jof-07-00925-t003:** Characteristics of IFI.

IFI	Study Populations (*n* = 195)
All IFI (Proven, Probable and Possible)	67 (34%)
All PP-IFI	25 (12.8%)
Proven IFI	9 (5%)
Proven Candidiasis	4
Proven Mucormycosis	4
Proven Cryptococcosis	1
Probable IFI	
Probable pulmonary Aspergillosis	16 (8%)
PP-IFI within 100 days of HCT	10 (5.1%)
Probable Aspergillosis	6
Proven Candidasis	3
Proven Cryptococcosis	1
PP-IFI within 180 days of HCT	13 (6.6%)
Probable Aspergillosis	8
Proven Candidiasis	3
Proven Cryptococcosis	1
Proven Mucomycosis	1
Median days to PP-IFI with 180 days of HCT	40 (range, 11–165)

Abbreviations: IFI: invasive fungal infection, HCT: hematopoietic stem cell transplant, PP: proven and probable.

**Table 4 jof-07-00925-t004:** Univariate analysis of clinical factors and IFI.

Variable	Number of Patients	Nb of Patients with IFI (%)	*p* Value
**Age**			0.23
≥65	183	61 (33)	
<65	12	6 (50)	
**Sex**			0.13
Male	123	47 (38)	
Female	72	20 (27)	
**Disease**			0.45
AML	115	37 (32)	
ALL	29	8 (27)	
Lymphoma	31	13 (42)	
Others	20	9 (45)	
**Disease status before**			**0.023**
HCT			
CR	118	33 (30)	
SD	9	5 (55)	
Refractory	19	11 (58)	
Others	35	15 (43)	
**Donor type**			**0.0003**
MRD	114	27 (23)	
MMRD	81	39 (48)	
**Conditioning regimen**			0.83
MAC	153	52 (35)	
RIC	42	15 (36)	
ATG			0.082
Yes	170	54 (32)	
No	25	13 (52)	
**GvHD prophylaxis**			**0.003**
CsA	94	23 (24)	
CsA-MMF	17	4 (24)	
CsA-MMF-PTCy	84	40 (48)	
**aGvHD**			0.16
Yes	50	15 (30)	
No	145	52 (36)	

Abbreviations: AML: acute myeloid leukemia, ALL: acute lymphocytic leukemia, HCT: hematopoietic stem cell transplant, B63Dversus-host disease, CsA: cyclosporine, MMF: mycophenolate mofetil, PTCy: post-transplant cyclophosphamide, ATG: anti-thymocyte globulin.

## Data Availability

The data presented in this study are available on request from the corresponding author. The data are not publicly available due to local IRB restrictions.

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
