# Peer review of "The Use of Voriconazole as Primary Prophylaxis for Invasive Fungal Infections in Patients Undergoing Allogeneic Stem Cell Transplantation: A Single Center’s Experience"

_jof, 2021, doi:10.3390/jof7110925_

Round 1

Reviewer 1 Report

This paper discusses the role of voriconazole as primary prophylaxis for IFI in patients undergoing allogeneic stem cell transplantation in a single Bone Marrow Transplant Unit. Data were collected retrospectively.

Some comments for the authors:

1) Data were collected over a period of 6 years. Have environmental risk factor for IFI changed during this period?

2) Voriconazole oral formulation is safe and well tolerated. Why was voriconazole administered only intravenously throughout hospitalization?

3) How many patients had an history of IFI before allogeneic stem cell transplant?

4) Could you please detail the incidence of aGVHD grade II-IV and grade I?

5) Are risk factors for IFI different at different timepoint after transplant?

6) Data from GITMO (Girmenia at el, 2014) demonstrated that aGvHD and cGvHD are independent risk factors for the development of IFI. Can you please comment more on the role of GvHD in IFI development within your cohort?

7) Could you please better detail the characteristics of PP-IFI and the time to onset after transplant? (Tab. 3)

8) Are you planning to introduce TDM for voriconazole in your population? Can you please comment on the use of TDM for voriconazole?

Author Response

Response to Reviewer 1 comments

Point 1: Data were collected over a period of 6 years. Have environmental risk factors for IFI changed during this period?

Answer: Over the past 6 years, the environmental risk factors for development of invasive infections have not changed. Knowing that environmental risk factors include type of ventilation, type of medical care and residence of patients, these factors were almost the same over the past 6 years.

Point 2: Voriconazole oral formulation is safe and well tolerated. Why was voriconazole administered only intravenously throughout hospitalization?

Answer: In the first few days post-transplant, patients experience severe nausea, vomiting and decreased oral intake due to the intensive chemotherapy given. This is in addition to severe oral mucositis secondary to neutropenia. In this period of time, the goal is to minimize oral medications and for this reason Voriconazole is given as IV then changed to oral when patients tolerate oral intake on discharge.

Point 3: How many patients had an history of IFI before allogeneic stem cell transplant?

Answer: Patients with IFI prior to transplant were not included in this study because they were already receiving treatment for IFI or receiving another anti-fungal mainly Posaconazole as secondary prophylaxis. Only 4 patients had IFI prior to transplant.

Point 4: Could you please detail the incidence of aGVHD grade II-IV and grade I?

Answer: A detailed characteristic of aGVHD will be included in the manuscript including site of GVHD and requirement of steroids.

Point 5: Are risk factors for IFI different at different timepoint after transplant?

Answer: Risk factors for IFI are mainly GVHD, prolonged neutropenia, and older age. These factors were not different at different timepoint. At different timepoint after transplant, there was heterogeneity in terms of age, incidence of GVHD and white blood cell count over time.

Point 6: Data from GITMO (Girmenia at el, 2014) demonstrated that aGvHD and cGvHD are independent risk factors for the development of IFI. Can you please comment more on the role of GvHD in IFI development within your cohort?

Answer: Of course, this will be included in our revised manuscript submission.

Point 7: Could you please better detail the characteristics of PP-IFI and the time to onset after transplant? (Tab. 3)

Answer: In the text, mainly the result section, the characteristics of all PP-IFI were detailed in terms of pathogens and detection. If needed to be included in the table 3, it can be definitely added.

Point 8: Are you planning to introduce TDM for voriconazole in your population? Can you please comment on the use of TDM for voriconazole?

Answer: Unfortunately, Therapeutic Drug Monitoring is not feasible currently at our institution. We are working to introduce it as soon as possible. The use of TDM of voriconazole will be commented on in the revised manuscript.

Reviewer 2 Report

“The use of voriconazole as primary prophylaxis for invasive fungal infections in patients undergoing allogeneic stem cell transplantation: A single center experience”

The report could be interesting if the data are clarified as far as the number of PP infections; the percentages of proven could be important data.

Abstract and everywhere else, no Italics for the term “species”.

Describe what is AML?

The authors need to report the data as MSG has done “(proven and probable 8% to 12%)” has done since they are using that as their standard. The abstract states:

Lines 23-24. “The incidence of IFI including proven, probable and possible IFI was 34%. Proven-Probable (PP) IFI at any time of transplant was 12.8%”.

Perhaps the order of the tables could be inverted, summarize the result of prophylaxis in Table 1 and liste those by the MSG group using the same categories of IFI.

Line 65 information is well known, not needed.

Is Table 2 needed? Or that information presented to complement the overall incidence of IFI

Table 3. Somehow the results or their presentation don’t seem to match the abstract.

Is table 4 needed?

Were voriconazole MICs determined? It will be interesting to know if voriconazole MICs for the isolates from PTS with proven disease were high or above the available reference BPs or ECVs.

Under “conclusions” very low is not acceptable, need to provide percentages after the incidence of proven invasive FI is clarified. The Discussion is a bit too long.

Author Response

Response to Reviewer 2 comments

Point 1: The report could be interesting if the data are clarified as far as the number of PP infections; the percentages of proven could be important data.

Answer: Our report included the incidence of all invasive fungal infections including proven, probable and possible IFI. Table 3 summarizes the incidence of IFI at any time point of the study and the incidence of PP (Proven-Probable) IFI at day 100 and day 180. There were only 4 cases of proven mucormycosis in our cohort. In the result section, the incidence of proven mucormycosis was stated as 2% (4 cases). The incidence of proven IFI will be added to table 3.

Point 2: Abstract and everywhere else, no Italics for the term “species”.

Answer: Italics for the term “species” were all corrected.

Point 3: Describe what is AML?

Answer: The abbreviation AML in the abstract was corrected.

Point 4: The authors need to report the data as MSG has done “(proven and probable 8% to 12%)” has done since they are using that as their standard. The abstract states: Lines 23-24. “The incidence of IFI including proven, probable and possible IFI was 34%. Proven-Probable (PP) IFI at any time of transplant was 12.8%”.

Answer: The data in the abstract will be reported similarly to MSG data.

Point 5: Perhaps the order of the tables could be inverted, summarize the result of prophylaxis in Table 1 and liste those by the MSG group using the same categories of IFI.

Answer: Tables are listing by the following orders: Patients and disease characteristics, transplant complications, IFI characteristics and finally Univariate analysis of variables and incidence of IFI. In table 3, the result of prophylaxis will be listed by MSG group using same categories

Point 6: Line 65 information is well known, not needed.

Answer: Line 65 will be removed

Point 7: Is Table 2 needed? Or that information presented to complement the overall incidence of IFI

Answer: Table 2 summarizes the complications of transplant including infections and GvHD. As it is well established by multiple previous studies that patients with high risk (including GVHD and viral reactivation), these results were included to present the incidence of possible high risk patients. However, as there was no correlation between these complications and the incidence of IFI in our cohort, table 2 can be removed.

Point 8: Table 3. Somehow the results or their presentation don’t seem to match the abstract.

Answer: In table 3, we reported the incidence of all IFI, PP-IFI and PP-IFI at day + 100 and + 180. The incidence of IFI at day 100 was 5.1 % (10 cases). Between day + 100 and + 180, there were 3 additional cases making the total number at + 180 equal to 13 cases which is 6.6% of the study population.

In addition, the incidence of all IFI at any time after transplant was 34% (67 patients). From these, only 25 had proven-probable IFI which represents 12.8% of the study population.

The revised manuscript will address the presentation of these results.

Point 9: Is table 4 needed?

Answer: Table 4 summarizes the result of a univariate analysis of patients and disease characteristics and the incidence of IFI. In previous studies, multiple factors have been identified as high risk for the development of IFI post allogeneic stem cell transplant. In this table, we analyzed the relation between these factors and IFI to better identify the high risk population in our study.

Point 10: Were voriconazole MICs determined? It will be interesting to know if voriconazole MICs for the isolates from PTS with proven disease were high or above the available reference BPs or ECVs.

Answer: Unfortunately, It was only last year we started to work on the speciation and susceptibility of isolates at AUBMC. MICs were not determined in any of our patients.

Point 11: Under “conclusions” very low is not acceptable, need to provide percentages after the incidence of proven invasive FI is clarified. The Discussion is a bit too long

Answer: Conclusion will provide exact percentages and the discussion will be adjusted.

Reviewer 3 Report

It is widely accepted that recipients of alloHSCT should receive antifungal prophylaxis. Antifungals with anti-mold activity such as voriconazole should be considered, in particular, in high risk patients. However, the best prophylactic option agent of choice remains to be determined. Voriconazole as primary prophylaxis is given in most of the guidelines a B recommendation, after other agents with an A recommendation due to better evidence.

The authors present a retrospective study analyzing voriconazole as primary prophylaxis in HSCT. Although it is a single-center experience, the sample is large. The present study adds information to the efficacy and tolerability of voriconazole in a real-life setting, but does not clarify important pending issues (such as the direct comparison between voriconazole and posaconazole or isavuconazole, or the usefulness of using TDM to adjust dosage or cost-effectiveness analysis), so that it is not innovative in this sense

1.What proportion of the HSCT in your study could be considered “high risk” (as opposed to standard risk)?

  1. To clarify, please include the incidence rate of invasive aspergillosis in the abstract, on top of global IFI incidence.
  2. Please, give a reference of incidence of IFI in the setting of post-HSCT that allows you to consider your figures are low (a reference measured along a length of time similar to the one you use).
  3. In the introduction, comment on the current grade of recommendation of voriconazole in the setting of post-HSCT, as compared to the other agents.
  4. Reference to the Wingard trial (reference 4) page 2, line 50) is not did precise: the differences in IFI incidence with voriconazole as compared to fluconazole were not significant, please, clarify this statement.
  5. The statement referring to Marks study (reference 5, page 2 line 53) is not precise: the study showed superiority in a composite endpoint that included not only survival at day 180 post-transplant as well as proven or probable IFI during that same period, but also discontinuation of the study drug for [14 days) (Hoenigl). Please, clarify.
  6. State in the introduction the knowledge gaps you intend to clarify with your study.
  7. Please, provide the total number of IA (not only pulmonary)(page 4, line 135).
  8. Describe further what you considered as “Invasive candidiasis”: were the “probable candidiasis” hepatosplenic candidiasis? No candidemia? In what samples was Candida isolated?
  9. In page 4, line 138, it is not clear whether there were 1 or 4 mucormycoses. Please, clarify.
  10. Total PP IFI 10+13=25? When did the remaining 2 occur? Were the early IFIs (<100 days) pre or post-engraftment?
  11. Reference 8 in the discussion (page 8, line 190): Is 7.8% a figure reflecting incidence or prevalence? If it is incidence, specify the period of time. Did studies used as reference have a similar proportion of high risk patients? Were there other differential factors, apart from a different prophylactic agent, that could justify the higher incidence of IFI? Were the etiologies similar?
  12. Please, reference all the trials mentioned in the discussion next to their data (pages 8 and9): systematic review and meta-analysis, etc.

Author Response

Response to Reviewer 3 comments

  1. What proportion of the HSCT in your study could be considered “high risk” (as opposed to standard risk)?

Answer: Multiple factors have been identified as high risk for IFI in patients undergoing allo-HSCT mainly by GITMO. High risk factors for early IFI include grade III-IV acute GvHD, transplant from unrelated/mismatched donor in combination with grade II GvHD, steroid use, CMV reactivation, prolonged neutropenia and iron overload. Based on the available data from our study, patients with at least one high risk classifying factor were as follows: grade III-IV GvHD (14 patients: 7%), Mismatched related donor (82 patients: 42%), steroid use (45 patients: 23%), CMV reactivation (90 patients: 46%).  Other factors for late IFI include refractory or recurrent disease at the time of transplant which represents 19 patients (10%) of our study population. In table 4, a univariate analysis for any of these factors and the incidence of IFI was done to identify high risk patients in our study.

  1. To clarify, please include the incidence rate of invasive aspergillosis in the abstract, on top of global IFI incidence.

Answer: This will be addressed in the abstract.

  1. Please, give a reference of incidence of IFI in the setting of post-HSCT that allows you to consider your figures are low (a reference measured along a length of time similar to the one you use).

Answer: Voriconazole as primary prophylaxis in allo-HCT patients was mainly studied in 2 clinical trials. The first trial in 2010 by Wingard et al, compared voriconazole and itraconazole in this setting. It was followed by another trial by Marks et al in 2011 comparing voriconazole to fluconazole. Our revised manuscript will compare our data to these studies.

  1. In the introduction, comment on the current grade of recommendation of voriconazole in the setting of post-HSCT, as compared to the other agents.

Answer: There are many recommendations from different societies on the use of antifungals in the setting of allo-HSCT. We will include the most important and use ones in the revised manuscript.

  1. Reference to the Wingard trial (reference 4) page 2, line 50) is not did precise: the differences in IFI incidence with voriconazole as compared to fluconazole were not significant, please, clarify this statement.

Answer: In this  double-blind, randomized trial comparing prophylactic voriconazole (n = 305) with fluconazole (n = 295) given for 100 days post-alloHSCT, results no significant difference in the primary endpoint of fungal-free survival at 180 days (hazard ratio [95% CI]: 1.07 [0.82–1.40]); however, in a subset of patients receiving alloHSCT for therapy of acute myeloid leukemia, fewer IFIs were observed with voriconazole prophylaxis than with fluconazole (hazard ratio [95% CI]: 2.03 [1.29–3.19]).

  1. The statement referring to Marks study (reference 5, page 2 line 53) is not precise: the study showed superiority in a composite endpoint that included not only survival at day 180 post-transplant as well as proven or probable IFI during that same period, but also discontinuation of the study drug for [14 days) (Hoenigl). Please, clarify.

Answer: This trial is an open-label, randomized trial comparing voriconazole (n = 234) and itraconazole (n = 255) in 489 recipients of alloHSCT following treatment for 100–180 days [30]. The primary objective was a composite endpoint incorporating survival without proven/probable IFI at 180 days post-alloHSCT transplant and ability to tolerate therapy for at least 100 days with fewer than 14 days’ interruption. The voriconazole arm met the criteria for superiority in the primary endpoint when compared with the itraconazole arm (48.7% vs. 33.2%, respectively; p < 0.01). This will be clarified in the introduction.

  1. State in the introduction the knowledge gaps you intend to clarify with your study.

Answer: This point will be addressed in the introduction.

  1. Please, provide the total number of IA (not only pulmonary) (page 4, line 135).

Answer: In our study, we had only pulmonary aspergillosis. We identified 16 cases of pulmonary aspergillosis at any time after transplant. This was reported in the results, section incidence and characteristics of IFI.

  1. Describe further what you considered as “Invasive candidiasis”: were the “probable candidiasis” hepatosplenic candidiasis? No candidemia? In what samples was Candida isolated?

Answer: The 4 cases of candida were candidemia with candida found in blood cultures. These are considered proven candidiasis. Proven was mistakenly replaced by probable in table 3. The table is corrected in the revised manuscript.

  1. In page 4, line 138, it is not clear whether there were 1 or 4 mucormycoses. Please, clarify.

Answer: This information will be clarified in the revised manuscript.

  1. Total PP IFI 10+13=25? When did the remaining 2 occur? Were the early IFIs (< 100 days) pre or post-engraftment?

Answer: The total number of patients with IFI was 25. 10 cases were identified before day + 100 and 13 cases were identified before day + 180 (10 of them are the same patients identified before day + 100). That being said, we have only 3 cases identified between day + 100 and day + 180. The remaining cases are identified after day + 180. The revised manuscript will include the relation between engraftment and time of IFI before day 100.

  1. Reference 8 in the discussion (page 8, line 190): Is 7.8% a figure reflecting incidence or prevalence? If it is incidence, specify the period of time. Did studies used as reference have a similar proportion of high risk patients? Were there other differential factors, apart from a different prophylactic agent, that could justify the higher incidence of IFI? Were the etiologies similar?

Answer: In reference 8, the incidence of IFI in allo-HCT patients was 7.8% during the 5-year study period. In another prospective study on invasive fungal infections between 2001 and 2006, the cumulative incidence was highest for aspergillosis, followed by candidiasis. We will add this reference in our manuscript in addition to comparison of different factors.  

  1. Please, reference all the trials mentioned in the discussion next to their data (pages 8 and9): systematic review and meta-analysis, etc.

Answer: This will be addressed in the revised manuscript.

  1. Compare and comment the voriconazole discontinuation rate of the present study to others (Reasons for voriconazole prophylaxis discontinuation in allogeneic hematopoietic cell transplant recipients: A real-life paradigm. Chan SY et al, Med Mycol 2020)

Answer: A comparison of the discontinuation rate will be added to the manuscript.

  1. What does the paragraph on isavuconazole (page 9, lines 257-264) add? Similar efficacy? Consider eliminating this paragraph.

Answer: This paragraph was added to compare the rate of discontinuation between voriconazole and isavucinazole. However, the rate of discontinuation and reasons will be replaced by the above cited reference.

  1. Were all the patients that discontinued voriconazole given anidulafungin? Did any of these present an IFI? We suggest eliminating the sentence about micafungin (page 9, line 269-271), as it does not add pertaining information.

Answer: The sentence about micafungin will be eliminated. Most of patients received anidulafungin after the discontinuation of voriconazole (16 patients). Two out of 16 patients had IFI (1 mucormycosis and 1 pulmonary IA).

  1. Add to limitations: NON-comparative study, possible loss of follow-up as a retrospective study, does not clarify important pending issues such as the direct comparison between voriconazole and posaconazole (or isavuconazole), the usefulness of using TDM to adjust dosage or cost-effectiveness single center

Answer: Our study is a retrospective study on the incidence of IFI in patients receiving voriconazole as primary prophylaxis for IFI in allo-HSCT patients. These patients are closely followed up by our transplant team up to 1 year after the date of transplant. Knowing that our study does not compare the efficacy of voriconazole to other antifungals, we showed a low incidence of IFI in this population using voriconazole. The use of TDM is not currently available in our centre, we are currently working to introduce it.

  1. Comment on the advantages you consider voriconazole could offer over the alternative options.

Answer: Our study demonstrated a low incidence of IFI at day 100 and day 180 using voriconazole in patients who underwent allo-HSCT for hematological malignancies. We will add the suggested advantage of using voriconazole based on our reported data.

Round 2

Reviewer 3 Report

I still believe that some of the limitations of the present study (such as the direct comparison between voriconazole and posaconazole (or isavuconazole), the usefulness of using TDM to adjust dosage or cost-effectiveness) could be expressed as a need for further studies. This does not undermine the contribution of the present study, but highlights the remaining knowledge gaps